# Graph-Based Analyses of Dynamic Water-Mediated Hydrogen-Bond Networks in Phosphatidylserine: Cholesterol Membranes

**DOI:** 10.3390/biom13081238

**Published:** 2023-08-11

**Authors:** Honey Jain, Konstantina Karathanou, Ana-Nicoleta Bondar

**Affiliations:** 1Faculty of Physics, University of Bucharest, Atomiștilor 405, 077125 Măgurele, Romania; 2Department of Physics, Freie Universität Berlin, Arnimallee 14, 14195 Berlin, Germany; 3IAS-5/INM-9, Forschungszentrum Jülich, Institute of Computational Biomedicine, Wilhelm-Johnen Straße, 52428 Jülich, Germany

**Keywords:** depth-first-search algorithm, lipid–water hydrogen-bond network, hydrogen-bond network topology, POPS-cholesterol hydrogen-bond networks

## Abstract

Phosphatidylserine lipids are anionic molecules present in eukaryotic plasma membranes, where they have essential physiological roles. The altered distribution of phosphatidylserine in cells such as apoptotic cancer cells, which, unlike healthy cells, expose phosphatidylserine, is of direct interest for the development of biomarkers. We present here applications of a recently implemented Depth-First-Search graph algorithm to dissect the dynamics of transient water-mediated lipid clusters at the interface of a model bilayer composed of 1-palmytoyl-2-oleoyl-sn-glycero-2-phosphatidylserine (POPS) and cholesterol. Relative to a reference POPS bilayer without cholesterol, in the POPS:cholesterol bilayer there is a somewhat less frequent sampling of relatively complex and extended water-mediated hydrogen-bond networks of POPS headgroups. The analysis protocol used here is more generally applicable to other lipid:cholesterol bilayers.

## 1. Introduction

Hydrogen (H)-bonds and H-bond networks shape the structure and dynamics of lipid–lipid interactions [1] and the fluidity of the lipid bilayer [2,3,4]. H-bonding properties depend on the nature of the lipid headgroup. The anionic phosphatidylserine (PS) has more inter-lipid H-bonds than, e.g., the zwitterionic phosphatidylethanolamine (PE) and phosphatidylcholine (PC), or than the anionic phosphatidylglycerol (PG) and phosphatidic acid (PA) [5,6,7]. This propensity for intermolecular H-bonding of PS may be associated with somewhat larger bilayer thicknesses [7]. 

Lipid clusters can be defined as groups of lipids H-bonded via direct H-bonds, water molecules, or ion interactions. The clustering of PS appears to be associated with the formation of nanometer-sized domains [8,9,10] that could provide, e.g., a platform for peripheral protein recruitment [11], viral assembly [12], and the binding to the membrane of the influenza A virus matrix protein [9]. Altered distribution of PS has been associated with a number of human diseases [13], including cancer [14,15,16]. The formation of lipid clusters, particularly in the case of PS [17,18,19], PG [17], and phosphatidylinositol 4,5-bisphosphate (PIP2) [20], might be promoted by the presence of cations. Experiments using, e.g., time-resolved fluorescence spectroscopy [18], fluorescence correlation spectroscopy [20,21], steady-state probe-partitioning fluorescence resonance energy transfer (SP-FRET) [20], and molecular dynamics (MD) simulations [18,19,22] suggested that the presence of sodium [9,18,21,23,24,25,26], potassium [18,22], calcium [19,22,25,26,27,28], or magnesium ions [28] may be associated with the clustering of lipids. Details of ion-mediated lipid cluster formation depend both on the cation and on the lipid headgroup. For example, compared to potassium, the presence of sodium ions tends to be associated with larger clusters in PIP2 bilayers [22] and shows a higher affinity to a membrane composed of 1-palmytoyl-2-oleoyl-*sn*-glycero-2-phosphatidylcholine (POPC) and 1-palmytoyl-2-oleoyl-*sn*-glycero-2-phosphatidylserine (POPS) [18]. At the interface of a POPS/POPC bilayer, sodium ions have reduced mobility relative to POPC [18]. Likewise, sodium and potassium ions have weaker binding at a PC than a PE lipid membrane interface [29].

Cholesterol favors interactions with anionic lipids (PS and PG) as compared to the zwitterionic PC and PE [30,31,32,33], and it can H-bond to lipid headgroups [34]. Cholesterol prefers H-bond to single phospholipids [35], typically with the lipid carbonyl or phosphate oxygen atoms [35,36,37]. Intercalation of cholesterol molecules could influence the packing of anionic lipids [30], looser at the lipid headgroups and tighter at the tails [38] and be associated with reduced sodium ion binding at the interface [39]. 

MD studies have shown that lipids can form H-bond networks mediated by direct or water-mediated H-bonds between lipid headgroups [5,17]. The water dynamics at the lipid interface is slower than the bulk [1,40,41]. The slower relaxation rates of interfacial waters H-bonded to lipids suggest that H-bond networks in the hydration layer facilitate lipid–lipid interactions [1]. The number of H-bonds between water molecules at the interface of DMPC [1] and DOPS [42] bilayers decreases with an increasing membrane depth.

To identify and characterize dynamic water-mediated clusters of lipid headgroups, we have recently implemented a Depth-First-Search (DFS)-based graph algorithm that uses Connected Component searches to explore the nodes of the H-bond graphs and identify four main types of lipid H-bond clusters [17]. Clusters are then characterized according to properties such as cluster size, which is given by the number of nodes (lipid headgroups) that constitute the cluster, and cluster length, which is given by the number of H-bonds within the longest H-bond path within the cluster [17]. Using the DFS-based algorithm we implemented, we dissected the water-mediated H-bond clusters at the interface of hydrated lipid bilayers composed of POPE, POPG, 3:1 POPE:POPG, 5:1 POPE:POPG, POPS, and an *Escherichia coli* lipid membrane model, all without cholesterol [17]. We found that, regardless of the lipid membrane composition, short linear H-bond arrangements, typically two lipid headgroups that H-bond directly or via one water molecule, were sampled frequently during atomistic MD simulations [17]; relative to the other lipids studied, POPS could engage more frequently in circular arrangements of three lipid headgroups [17]. 

Here, we apply the DFS graph algorithm to characterize dynamic water-mediated interactions in a POPS:cholesterol membrane with 10% cholesterol, which is in the range of cholesterol concentrations relevant to the plasma membrane of a healthy eukaryotic cell [43]. As a reference, we use a hydrated POPS bilayer without cholesterol. We compute separately H-bond clusters contributed by phosphate, serine vs. ester groups; likewise, we compute separately H-bond clusters that involve direct POPS:POPS H-bonds, vs. one-water bridges or ion-mediated bridges between POPS headgroups. We find that cholesterol disfavors the transient formation of H-bonded POPS clusters. 

## 2. Methods

### 2.1. MD Simulations of Hydrated Lipid Membranes

CHARMM-GUI [44,45,46] was used to generate coordinates for two distinct hydrated POPS bilayers (Table 1): (i) without cholesterol; (ii) with 10% cholesterol. All simulation systems have 0.15 M neutralizing KCl salt and a starting size of approximately 81 Å × 81 Å × 85 Å.

We used the CHARMM36 force field for lipids and ions [47,48,49,50,51,52] and the TIP3P water model [53]. All simulations were performed with NAMD 2.13 [54,55]. The standard CHARMM-GUI scheme for the initial equilibration was used. For the production runs, we used a Langevin dynamics scheme [56,57] with an oscillation period of 200 fs, a damping time scale of 100 fs, and a damping coefficient of 5 ps^−1^ in the *NPT* ensemble (constant number of particles *N*, constant pressure *P* = 1 bar, and constant temperature *T* = 310.15 K). We used the smooth particle mesh Ewald summation to compute Coulomb interactions [58,59] and a 10–12 Å switch function for real-space interactions; covalent bonds to H-atoms were fixed. An integration step of 1 fs was used throughout equilibration and the first 1 ns of the production runs; for the production runs, we used a multiple time-step integration scheme [60,61] with 1 fs for bonded interactions and 2 fs for short-range nonbonded forces and long-range electrostatics. Coordinates were saved every 10 ps for data analyses. The total sampling time of the two simulations was 1 μs.

### 2.2. H-bond Criteria, H-bond Graphs, H-bond Paths, and H-bond Occupancies

We identified H-bonds using a distance criterion whereby two groups were considered as H-bonded if the distance between the H-atom and acceptor hetero-atom was within 2.5 Å. This single distance criterion gives results largely equivalent to a combined distance and angle criterion of H-bond distances ≤3.5 Å between the H-bond donor and acceptor hetero-atoms, and H-bond angles within 60° [62]. For potassium ion-mediated bridges between two POPS phosphate groups, we used a distance criterion of 4 Å. 

A H-bond graph consists of nodes, which here are the POPS or cholesterol headgroups, and the edges, which are direct, 1 water-mediated, or ion-mediated H-bond connections between the nodes. A H-bond path between two lipid headgroups connects these two headgroups via the shortest distance H-bond path, i.e., via the smallest number of intermediate H-bonds. The occupancy of a H-bond is the percentage of coordinate sets utilized in analyses in which the H-bond criterion is satisfied [24,63]. We used Bridge [63,64] to compute H-bond occupancies. 

### 2.3. H-bond Clusters, Topology, Cluster Size, Path Length, and Occupancy of Lipid Clusters 

A H-bond cluster consists of nodes and edges that interconnect with each other. The topology of a H-bond lipid cluster is defined as the geometric arrangement of the nodes and edges in that cluster. To identify lipid clusters, we used the recently developed DFS algorithm [17] that analyses the H-bond graph to identify transient lipid clusters and the topologies of these clusters. Briefly, the DFS algorithm treats the nodes and edges of the H-bond graph as, respectively, data points and relationships between these data points. The algorithm explores all graph nodes, monitors H-bond distances in the coordinate dataset (MD trajectory), and constructs adjacency or connection matrices that represent the H-bond connections between pairs of nodes. The nodes of the graph are clustered, according to the H-bonding relationships between them, into four main topologies—linear, star and linear, circular, and complex combinations of these three arrangements. A lipid cluster is then characterized by its size, which is given by the number of lipid headgroups that constitute the cluster, and by the path length *L*, which is given by the number of edges (H-bonds) in the longest path connecting two nodes of the H-bond cluster.

The occupancy of a lipid cluster is the percentage of coordinate sets utilized in analyses in which the cluster is present. The length of the water wire that connects two lipid headgroups is given by the number of bridging H-bonded water molecules [63]; for simplicity, and because water bridges between lipids tend to have very small occupancies, here we considered only one-water bridges. All topology calculations were performed using the VMD [65] and MATLAB data analysis scripts of the DFS algorithm [17] as deposited in the Mendeley repository [66]. For data analyses, the complete trajectories of the production runs were used. 

### 2.4. Structure Factors and the Number of H-bonds per POPS

We used the MEMBPLUGIN [67] in VMD [65] to evaluate the average thickness of each bilayer, area per lipid, and order parameters. The H-bonds per lipid are reported as the average number of H-bonds sampled by the specific lipid group in each coordinate set.

All molecular graphics were prepared using VMD 1.9.4 [65]. Boxplots of H-bond occupancy values extracted with Bridge [63,64] were prepared using Python. Files pertaining to the simulations reported here are released as a public Mendeley repository (see the Data Availability Statement).

## 3. Results and Discussion

The simulations reported here have average membrane thickness, area per lipid, and order parameter values close to previous experiments and simulations. For the POPS membrane without cholesterol, the average membrane thickness we calculate is 41.2 ± 0.5 Å, which is close to values reported from previous MD simulations, e.g., 43.2 ± 0.5 Å [68], 40.6 Å [69], 42.4 ± 0.2 Å [70], 42.1 ± 0.5 Å [17], and to the 42.2 Å thickness measured by neutron and X-ray scattering in the presence of NaCl [69]. In the presence of 10% cholesterol, the POPS membrane is ~1.5 Å thicker (Table 1, Figure 1A,C), which is compatible with the ~2.2 Å increase, relative to pure DMPC, in the thickness of DMPC/cholesterol membranes with 10% cholesterol [71]. The calculated area per lipid for the POPS membrane is 59.8 ± 1.9 Å^2^, which is relatively close to previously reported values from MD simulations (58.4 Å^2^ [72], 57.5 ± 1.2 Å^2^ [70], and 62.0 Å^2^ [69]). A decrease of ~5 Å^2^ in the area per lipid and an increase in the order parameters found here in the presence of cholesterol (Figure 1B,D) are in reasonable qualitative agreement with the decrease by ~6–7 Å^2^ [73,74] in the area per lipid of a DOPC/cholesterol membrane with 10% cholesterol. Likewise, the order parameters for the POPS membrane (Figure 1D) are compatible with previous studies [70,72].

### 3.1. Direct and One-Water-Mediated H-bonding of POPS Headgroups

On average, each POPS headgroup has 1.5 ± 0.1 H-bonds with another POPS (Figure 2A), which is close to the 1.1–1.2 H-bonds/POPS headgroup reported previously from CHARMM36 MD simulations [7]. Each POPS headgroup also has, on average, 2.3 ± 0.2 H-bonds from one-water bridges (Figure 2B) and 1.5 ± 0.1 potassium ion interactions (Figure 2C). Each phosphate and serine group has about 1.3–1.4 H-bonds to a bridging water molecule (Figure 2), slightly higher than the ~1.1 one-water-mediated H-bonds per ester group (Figure 2B). When the ester groups are included in the H-bond computation, the average number of one-water-mediated H-bonds increases from 1.9 ± 0.1 (when only the POPS phosphate and serine groups are included in the H-bond computation) to 2.3 ± 0.2 (Figure 2), but these ester H-bonds have very low occupancies (Figure 3). The total average number of H-bonds per POPS is similar in membranes with and without cholesterol (Figure 2).

Overall, both the direct and one-water-mediated bridges between POPS have small occupancies: 83–93% of the direct H-bonds have occupancies within 10–15% (Figure 3A,C), and 69–100% of the one-water bridges computed for the different POPS H-bonding groups have occupancies within about 5% (Figure 3B,D), suggesting these interactions are quite dynamic. In the distribution of the occupancies of direct H-bonds and one-water-mediated bridges, there are also outliers: for each of the H-bonds and water-mediated bridges presented in Figure 3, about 4–8% of the data points are outliers, i.e., these are H-bonds and water-mediated bridges with occupancy values significantly larger than the average (Figure 3). The highest-occupancy H-bonds are, in the POPS membrane, a pair of POPS headgroups that remain within H-bond distance for 306.2ns (i.e., 61.2% POPS simulation) and, in the POPS:cholesterol simulation, a pair of POPS headgroups within H-bond distance for 360.2 ns (72% of the simulation). 

### 3.2. Water-Mediated H-bond Clusters in the POPS:Cholesterol Membrane

We used the DFS algorithm as illustrated in Figure 1 to characterize POPS clusters in membranes with cholesterol, using as a reference the results obtained for the POPS membrane without cholesterol [17,66].

We first considered all unique H-bonds sampled during the simulations. At any given time, about 15–18 clusters are likely to be sampled in the POPS:cholesterol membrane, that is, because each leaflet has 109 lipids (Table 1), at any moment of time about 81% of the lipids participate in dynamic, low-occupancy H-bond paths of at least two lipids. About half of these H-bond paths (8–9) are linear (Figure 1, Figure 4A). Linear, relatively short H-bond paths are favoured by POPS irrespective of whether cholesterol is present or not (Figure 4A and Figure 5) and are exclusively sampled in cholesterol clusters (Figure 6). Circular and star and linear clusters are rare and, when sampled, have minimal sizes (3 and 4–5 lipids, respectively, Figure 4A,B). Star and linear and circular arrangements may also be sampled transiently, such that, overall, most of the H-bond paths sampled are either linear or a more complex topology that includes a linear path segment (Figure 4).

As reported previously for a hydrated POPS bilayer [17], linear H-bond paths (Figure 1) mediated by one water molecule are sampled throughout the entire POPS simulation: on average, at least one linear path is sampled at any moment in time (Figure 5E). The one-water-mediated linear H-bond paths tend to be relatively short, composed of 2–3 lipids, that is, these are singular H-bonds, or two H-bonds with one common POPS headgroup (Figure 5E). These results for the reference POPS simulation are in qualitatively good agreement with our previous computations on hydrated POPS [17] and 4:1 POPC:POPG membranes [23]. In the presence of cholesterol, direct singular H-bonds (H-bond paths with length L = 1) between POPS headgroups are slightly less likely than in the POPS membrane (Figure 5A), though such paths are sampled throughout the entire simulation (Figure 5F). Taken together with the H-bond occupancy analysis summarized in Figure 3, the time series of the H-bond paths of different lengths (Figure 5) suggest that, overall, most H-bond clusters at the POPS headgroup interface are short and transient. It should also be noted that the exact number of H-bond paths, or clusters, sampled throughout the simulation trajectories (Figure 5), depends on the size of the lipid bilayer, and the time series presented in Figure 5 are meant as an illustration of the H-bond dynamics at the lipid headgroup interface of the hydrated membrane patches used here (Table 1).

As noted before for a hydrated POPS bilayer [17], linear paths with more than 2–3 lipids are sampled but only very rarely (Figure 4 and Figure 5A,B). Most (~60%) of the linear H-bond clusters sampled by POPS headgroups have a path length L = 1, and a significant percentage (25%) of the linear clusters have a path length L = 2. That is, when sampled, linear H-bond paths are typically short, only 2–3 lipids; longer linear paths with more lipids may be sampled but very rarely (Figure 5A,B). Cholesterol molecules may, infrequently, sample small clusters (L = 1, i.e., two cholesterol molecules); longer linear H-bond clusters with 3–4 cholesterol molecules H-bonded via one-water bridges are very rarely sampled (Figure 6). The infrequent sampling of cholesterol clusters can be attributed to its preference for H-bonding to single phospholipids [35]. Even in 1:1 concentrations, cholesterol–lipid is more likely than cholesterol–cholesterol interactions [35,36,37]. We also note that the sampling frequency of specific clusters can depend on the length of the simulation. 

## 4. Conclusions

Anionic PS lipid headgroups have a high propensity to H-bond [5,6,7], and topology analyses have revealed preferred arrangements of POPS bridged by water molecules [17]. Because the H-bond dynamics of PS can be affected by cholesterol [30,31,32,34], here we studied the dynamics of H-bond clusters sampled in atomistic simulations of a POPS membrane with cholesterol and compared the results with those for a cholesterol-free POPS bilayer. 

The concentration of cholesterol used here, 10%, approximates that of a healthy eukaryotic plasma membrane [43]. We stress, however, that the POPS:cholesterol membrane studied here is but a model membrane, because the eukaryotic plasma membrane is a complex mixture of no fewer than 11 different types of lipids [43]. As anticipated based on previous results reported in the literature [75], the POPS membrane is about 1.5 Å thicker when cholesterol is present (Figure 1A,C). 

Regardless of the presence of cholesterol, POPS samples dynamic (low-occupancy) water-mediated H-bond networks (Figure 3). As reported before, linear H-bond clusters are sampled more frequently than more complex topologies (Figure 1, Figure 4), yet more complex arrangements may be sampled, particularly in the absence of cholesterol (Figure 4). 

Regardless of the presence of cholesterol, POPS tends to form small linear lipid clusters with 2–3 lipids directly H-bonded or one-water-bridged (Figure 4 and Figure 5). Cholesterol is associated with fewer POPS being part of complex H-bond clusters (Figure 4). 

POPS-cholesterol clusters are sampled less frequently than POPS-POPS clusters, and cholesterol–cholesterol clusters are rare (Figure 6). When sampled, cholesterol-only clusters are small, typically only two cholesterol molecules (Figure 6); this finding is compatible with the previous study [76] reporting a preference of cholesterol molecules to form dimers. POPS clusters mediated by potassium ions prefer a linear arrangement of 3–4 lipids, whereas the more complex topologies are much less frequent (Figure 4). This is compatible with previous work [17] indicating that sodium-mediated POPS lipid clusters tend to prefer linear topologies with two lipid clusters; we note, however, that the number of cation-mediated clusters sampled during the simulations, and the size of the clusters, might depend on the length of the simulations and on the size of the lipid membrane patch. 

We anticipate that the protocol used here to evaluate the H-bonding of a simple cholesterol-containing model bilayer could become useful to dissect dynamic H-bond clusters sampled in simulations of more complex lipid mixtures, including for water-mediated lipid headgroup H-bond paths of potential interest for lateral proton transfer along lipid membrane interfaces, and to evaluate how different cations influence the sampling of transient H-bond clusters, and the preferred topologies of these clusters, in hydrated membranes with distinct lipid composition.

## Data Availability

Input coordinate and protein structure files, and trajectory files with coordinate sets from the simulations reported here, are openly available at Jain, Honey (2023), “Graph-based analyses of dynamic water-mediated hydrogen-bond networks in phosphatidylserine:cholesterol membranes”, Mendeley Data, V1, doi: 10.17632/h8y6zjckcw.1 as accessed on 11 August 2023.

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
