# Peer review of "Graph-Based Analyses of Dynamic Water-Mediated Hydrogen-Bond Networks in Phosphatidylserine: Cholesterol Membranes"

_biomolecules, 2023, doi:10.3390/biom13081238_

Round 1

Reviewer 1 Report

This ms presents a graph theory analysis of clustering of simulations of pure POPS bilayers and those with 10% cholesterol in a KCl solution.   Clustering of POPS is a combination of direct H-bonding between lipids, water mediated, and ion mediated.  Cholesterol, which shows little self-clustering, tends to destabilize clustering of POPC.  The simulations and analysis are well done, and the conclusions are valuable to the field.  I recommend publication with minor revisions.

Scientific:

1.      It is stated that the systems were run with 150 mM KCl, as appropriate when modeling the inner leaflet of cells.  However, sodium ions were noted five times in the text (page 2, bottom of paragraph 2; page 4, line 3; page 5, last line of figure caption; page 12, twice in figure caption.  Please clarify.

2.      The authors might consider commenting on the possible effects of sodium and calcium on POPC clustering. A sizable body of experimental work on PIP2 and recent simulations show that these ions are dramatically better cluster formers than potassium.

3.      Page 3, first paragraph of Results and Discussion. The numerical units of surface area appear to be ok, but the units should be angstroms-squared.

Writing:

Small comment first. No need to write “POPC lipid molecule” (e.g., bottom page 3), and similar phases like “POPC lipids”.  The readers now that POPC is a lipid and a molecule.  “POPC” is plenty in most all cases.

Bigger comment: The Results section contains numerous dense figures and relatively little exposition. After reading the paper several times I still don’t have a good sense of the lifetimes of the clusters (the times series in Fig 6 suggests long lifetimes while Fig 9 suggests short), and exactly why cholesterol is destabilizing.  Consider a thorough rewrite of this section with a focus on the big picture.

Minor editing of English language required.

Reviewer 2 Report

In this article, the dynamics of a reference membrane made exclusively of POPS lipids and a model bilayer made of POPS lipids plus cholesterol are both studied by the author. According to the graph-based analysis, cholesterol decreases the likelihood that POPS lipids form rather extensive and complex water-mediated hydrogen-bond networks at the membrane interface. This shows that the dynamics of the lipid-water hydrogen-bond network are shaped by cholesterol, which may have an effect on how biomolecules and medicines interact at the membrane interface.

I recommend the major revisions of this article with the following comments.

1. What is POPS? Abstract section.

2. Sometimes you are using the author’s name and sometimes avoiding in the main body, I think you should opt for a single method and then keep on using a single method while referencing. You may check the method used in  Metric-based resolvability of polycyclic aromatic hydrocarbons with doi:10.1140/epjp/s13360-021-01399-8

3. There is no single symbol defined. You should put a table to define the parameters.

4. What are the results if graphs are connected or disconnected? Do we have a change in the results? See Figure 4,5,6.

5. There is no clear future direction stated.

6. There is no data regarding graph, networks, and how a molecular network is converted to a graph.

7. There are a few references that are not cited in the main body. Kindly see carefully examine and cite them.

8. Literature review is very short. I recommend adding more data try to add updated data only.

9. Sentence structure is quite weird, sometimes you are using non-italic then suddenly an italic word came up. You shouldn’t be using a non-italic italic combo in the same sentence.

10.   Reference 22 is cited more than once in a few consecutive sentences. You may write a single sentence and consecutive sentences then cite reference 22 one time only.

Moderate editing of English language required.

Round 2

Reviewer 2 Report

It seems authors changed a lot and addressed all the queries.

Minor editing of English language required